# *Aspergillus* Was the Dominant Genus Found during Diversity Tracking of Potentially Pathogenic Indoor Fungal Isolates

**DOI:** 10.3390/pathogens11101171

**Published:** 2022-10-11

**Authors:** Maria Andersson (Aino), András Varga, Raimo Mikkola, Camilla Vornanen-Winqvist, Johanna Salo, László Kredics, Sándor Kocsubé, Heidi Salonen

**Affiliations:** 1Department of Civil Engineering, Aalto University, FI-00076 Aalto, Finland; 2Department of Microbiology, Faculty of Science and Informatics, University of Szeged, Közép fasor 52, H-6726 Szeged, Hungary

**Keywords:** *Aspergillus*, *Chaetomium*, *Paecilomyces*, pathogenic indoor fungi, cleaning chemicals

## Abstract

Viable airborne pathogenic fungi represent a potential health hazard when exposing vulnerable persons in quantities exceeding their resilience. In this study, 284 indoor fungal isolates from a strain collection of indoor fungi were screened for pathogenic potential through the ability to grow in neutral pH at 37 °C and 30 °C. The isolates were collected from 20 locations including 14 problematic and 6 non-problematic ordinary buildings. Out of the screened isolates, 170 isolates were unable to grow at 37 °C, whereas 67 isolates growing at pH 7.2 at 37 °C were considered as potential opportunistic pathogens. Forty-seven isolates growing at 30 °C but not at 37 °C were considered as less likely pathogens. Out of these categories, 33 and 33 strains, respectively, were identified to the species level. The problematic buildings included known opportunistic pathogens: *Aspergillus calidoustus*, *Trichoderma longibrachiatum*, *Rhizopus arrhizus* and *Paecilomyces variotii*, as well as less likely pathogens: *Aspergillus versicolor*, *Chaetomium cochliodes*, *Chaetomium globosum* and *Chaetomium rectangulare.* Opportunistic pathogens such as *Aspergillus flavus, Aspergillus fumigatus*, *Aspergillus niger* and *Aspergillus tubingensis* and less likely pathogens such as *Aspergillus westerdijkiae, Chaetomium globosum* and *Dichotomopilus finlandicus* were isolated both from ordinary and from problematic buildings. *Aspergillus* was the dominant, most diverse genus found during screening for potentially pathogenic isolates in the indoor strain collection. Studies on *Aspergillus niger* and *Aspergillus calidodoustus* revealed that tolerance to cleaning chemicals may contribute to the adaptation of *Aspergillus* species to indoor environments.

## 1. Introduction

Invasive infections by opportunistic pathogenic fungi mostly affect immunocompromised persons. They represent a serious problem in buildings such as hospitals and nursing homes housing vulnerable persons [1,2,3,4]. Opportunistic pathogenic fungi represent a global emerging threat for COVID-19 patients, patients receiving immunosuppressive medication and for patients with uncontrolled diabetes [5,6,7,8,9,10,11,12,13].

Buildings represent hostile environments for fungi growing indoors. Indoor fungi have to develop tolerance towards environmental stresses. Stress tolerance is connected to virulence and emerging fungal pathogens may evolve in indoor environment [7]. Invasive infections by fungi from the genera *Aspergillus, Rhizopus* and *Trichoderma* are associated with renovations of mold-infested buildings [8,9,10,11,12]. Some *Aspergillus* and *Chaetomium* species unable to grow at 37 °C may cause onychomycoses and have been suspected to cause rare exotic infections [4,14,15].

Pathogenic fungi have been frequently screened for in hospitals [1,2,3,4,5], but the occurrence and identity of potentially pathogenic fungi in ordinary buildings not connected to indoor air quality complaints has not been investigated, nor has the effectiveness of cultivation-based methods for tracking diversity of viable pathogenic fungi. In this study we screened indoor fungal isolates for the ability to grow in neutral pH at 37 °C, considered as one of many prerequisites for pathogenicity [16]. This study is a continuation and a supplement of two congress abstracts concerning potentially pathogenic fungi in buildings [17,18].

## 2. Results

### 2.1. Fungal Isolates Screened Positive for Pathogenic Potential

Two hundred and eighty-four indoor fungal isolates from 20 buildings were screened for pathogenic potential by examining their ability to grow at 37 °C and 30 °C at neutral pH on tryptic soy agar (TSA). The results are presented in Table 1. The tested isolates were grouped into four categories, A–D: (A) isolates with high pathogenic potential, (B) isolates with moderate pathogenic potential, (C) isolates with less likely pathogenic potential and (D) isolates with no pathogenic potential. The results in Table 1 also show the diversity of the isolates identified to the genus level and their assignation to 26 morphotypes (MT1–MT26). The number of isolates in each morphotype is shown in Table 1.

Most of the 284 indoor isolates described in this study tested negative for pathogenic potential; 170 isolates were unable to grow at 37 °C. A total of 67 out of 284 indoor fungal isolates tested positive for pathogenic potential by growing at 37 °C in neutral pH. The 67 isolates were assigned to 13 morphotypes, 12 ascomycetous and one zygomycetous genus, respectively. The results in Table 1 also show that 54 isolates exhibited “high pathogenic potential” (category A isolates). These isolates represented 8 morphotypes (MT1–MT8) of the ascomycetous genera *Aspergillus* and *Trichoderma* and the zygomycetous genus *Rhizopus,* respectively. Thirteen isolates exhibited “moderate pathogenic potential” (category B isolates) and represented 5 morphotypes (MT9–MT13) of the ascomycetous genera *Aspergillus*, *Paecilomyces* and *Trichoderma*.

Isolates considered as exhibiting “less likely pathogenic potential” (Category C isolates) consisted of 47 isolates that grew at neutral pH at 30 °C but not at 37 °C. Twenty-four isolates represented two morphotypes of *Aspergillus* (MT14, MT15), twelve strains in each morphotype. Twenty-three isolates proved to be *Chaetomium*-like fungi representing four morphotypes (MT16–MT19).

The 170 isolates in category D, considered to have no pathogenic potential, were included in seven morphotypes (MT20–MT26). They were unable to grow at 37 °C in acidic or neutral pH, as the isolates of *Acrostalagmus*, *Penicillium* and *Rhizopus,* or even unable to grow on tryptic soy agar (TSA) plates at 30 °C, like the *Trichoderma* isolates.

### 2.2. Identification of Potentially Pathogenic Strains to the Species Level

Out of the 67 isolates of the 13 morphotypes (MT1–MT13), 34 strains fulfilling the above criteria for high or moderate potential pathogenicity were identified to the species level, and the selected toxins were identified. The identified strains, their origin, loci used for identification, NCBI GenBank accession numbers and selected toxins produced are collected in Table 2.

The species-level identification revealed species of the genera *Aspergillus, Rhizopus, Trichoderma* and *Paecilomyces.* The 33 selected strains of the six morphotypes (MT14–MT19) fulfilling the criteria of less likely pathogenic potential represented species of the genera *Aspergillus, Chaetomium* and *Dichotomopilus.*

Morphotypes MT1 and MT2 represented ophiobolin-producing toxigenic *Aspergillus calidoustus* strains. The four strains in MT1 exhibited yellow-beige colonies and produced Hülle cells, whereas the strain in MT2 exhibited grey colonies turning black after prolonged incubation, and no Hülle cells were observed. MT3 to MT5 represented *A. fumigatus, A. niger*, *A. pseudoglaucus* and *A. tubingensis*, producing yet unidentified toxic metabolites. Strains nontoxic in the two bioassays represented *A. flavus* isolates of morphotypes MT6 and MT9, exhibiting high and moderate pathogenic potential, respectively.

The genus *Paecilomyces* was represented by viriditoxin-producing *P. variotii* (MT10) and strains producing unidentified toxic metabolite, as well as non-toxic strains identified as *Paecilomyces* sp. (MT11–MT12). These strains in MT11 and MT12 were identical to each other by *CaM* sequence analysis, but it was not possible to assign them to any species yet. The toxigenic genus *Trichoderma* exhibited trilongin-producing *T. longibrachiatum* and *T. citrinoviride* screened positive for high and moderate pathogenic potential, respectively. The zygomycetous isolate was identified as *Rhizopus arrhizus* and proved non-toxic in the used bioassays.

The 33 selected toxigenic strains out of the 47 isolates able to grow at neutral pH at 30 °C (MT14–MT15) were identified as *Aspergillus westerdijkiae* producing stephacidin B, avrainvillamid and ochratoxin, and strains assigned to *Aspergillus* series *Versicolores* producing sterigmatocystin and averufin. Strains belonging to morphotypes MT16–MT19 were identified as chaetoglobosin-, chetomin- and chaetoviridin-producing *Chaetomium globosum,* and chaetomin- and chaetoviridin- producing *C. cochliodes*, as well as *C. rectangulare* and *Dichotomopilus finlandicus* producing unidentified toxic metabolite. 

### 2.3. Occurrence of Pathogenic Isolates in Moldy Problematic and Non-Problematic Ordinary Buildings

The 67 isolates out of the 284 isolates deposited in the strain collection characterized as opportunistic pathogens represented 26% of the deposited isolates. Forty-seven more isolates were characterized as less likely pathogens, representing 17% of the deposited isolates. These 114 pathogenic and less likely pathogenic fungal isolates representing 40% of the deposited strains emanated from 19 buildings, 14 problematic and 5 non-problematic. From one non-problematic building, only non-pathogenic isolates were deposited. The results are summarized in Table 3 and show that 19 out of 20 Finnish buildings contained culturable, potentially pathogenic fungi.

The genera *Aspergillus, Chaetomium, Trichoderma, Paecilomyces* and *Rhizopus* found in 19 buildings contained 11 known pathogenic and 6 less likely pathogenic species, respectively. Zygomycetous fungi as *Rhizopus* isolates were found in all buildings but potentially pathogenic isolates from only one problematic building. Representants of the genus *Penicillium* were also isolated from all buildings, but the strain collection contained no isolates fulfilling the pathogenic criteria.

Interestingly, potentially pathogenic isolates of *Aspergillus* sections Nigri, Circumdati and Fumigati and *Chaetomium*-like isolates like *C. globosum* and *D. finlandicus* were isolated from both problematic buildings and ordinary ones without reported moisture problems. The pathogenic isolates of *A. calidoustus, A. pseudoglaucus, A. versicolor*, *C. cochliodes*, *C. rectangulare*, *Trichoderma* and *Rhizopus* all emanated from problematic and moist buildings.

### 2.4. Characterization of Two Representants of the Major Pathogenic Genus: Aspergillus niger and Aspergillus calidodustus

Potentially pathogenic *Aspergillus* isolates represented the most numerous and diverse group among the potentially pathogenic isolates deposited in the strain collection. To elucidate the apparent adaptation of *Aspergillus* species to indoor environments, the effects of three chemicals (the tenside Genapol-X-80, the biocides borax and triclosan) used indoors were tested on the growth, competitiveness and differentiation of two *Aspergillus* strains: *A. niger* Asp21 and *A. calidoustus* MH34.

#### 2.4.1. Effect of Genapol-X-080 on the Competitiveness of the *Aspergillus niger* Strain Asp21 during 21 d of Incubation on Solid Culture Medium 

The effect of the non-ionic common synthetic tenside Genapol-X-080 on the competitiveness of an *A. niger* strain Asp 21 against *Paecilomyces* sp. strain Pec/skk and *Chaetomium cochliodes* strain CH2 was tested. The microtiter plates with wells inoculated with the three strains and imaged after 20 d of incubation are shown in Figure 1.

Figure 1 shows that the equal distribution of inoculated strains persisted over the course of 20 d in the plate without Genapol addition (Panel A), whereas the black conidia of strain Asp 21 occupied most of the wells in the plate where Genapol had been added (Panel B). Interpretations of the results in Figure 1 are summarized in Table 4 and Figure 2. The results in Table 4 show that in two plates with Genapol addition, the *A. niger* conidia occurred in 71%–88% of the 48 wells; *A. niger* in the control plate without Genapol occurred in 35% of the wells.

Figure 2 shows that out of the six wells in Figure 1 containing 9.4 µg ml^−1^, 4.7 µg mL^−1^, and 0 µg mL^−1^ of Genapol, 6, 5 and 2 wells, respectively, contained *A. niger* conidia. The results in Table 4 and Figure 2 indicated that in the plate containing Genapol addition, Asp 21 was able to infest wells previously inoculated with strains Pec/skk and CH2.

#### 2.4.2. Effects of Two Biocides and a Tenside on the Germination of Conidia, Resporulation and Production of Hülle Cells in an Exposed *Aspergillus calidoustus* Strain

The effects of two biocides, Borax and Triclosan, and the surfactant Genapol-X-080, on the germination of conidia and the formation of new conidia and Hülle cells was investigated. Spore suspensions of the *A. calidoustus* strain MH34 were exposed to the chemicals in malt extract broth and examined under light microscope after 3 d and 9 weeks of exposure. The results in Table 5 show that Triclosan was the most effective inhibitor of the germination of conidia; EC_100_ concentrations were 16 µg ml^−1^ for Triclosan and 5000 µg mL^−1^ for Borax. Genapol did not inhibit the germination of conidia in the tested concentration, the EC_100_ concentration was >5000 µg mL^−1^. Triclosan caused a strong Hülle cell formation at 8 µg mL^−^^1^, but no new conidiophores were seen. Genapol exposure in all the concentrations tested appeared to inhibit the formation of Hülle cells. The formation of new conidia was prevented by exposure to 1000 µg mL^−1^ of Genapol but occurred in amounts similar to that of the control in concentration of 100 µg ml^−1^ of Genapol.

## 3. Discussion

This study provides new information about the species diversity and potential pathogenicity of fungal species occurring in buildings. The genus *Aspergillus* was the dominant and most diverse representative of cultivable pathogenic indoor fungi. The adaptation to indoor environments exhibited by the pathogenic *Aspergillus* species may be connected to their tolerance to chemicals included in cleaning agents and building materials [19,27].

### 3.1. Detecting Indoor Fungi Exhibiting Pathogenic Potential by Screening for Growth in Neutral pH

In this study, 284 indoor fungal isolates from problematic and ordinary buildings were screened for pathogenic potential by analyzing their ability to grow at pH 7.2 at 37 °C and 30 °C. The majority, i.e., 170 of the 284 indoor isolates described in this study tested negative for pathogenic potential. Growth at 37 °C and neutral pH is considered as one of the prerequisites for fungal pathogenicity and ability to cause human infections [16]. It is also easy to test. The virulence of pathogenic fungi is connected to viability [8,12,28,29]. The majority of the metabolically active building mycobiota is cultivable, detectable and possible to be identified to the genus level by cultivation-based conventional methods [7,29,30].

Isolates characterized as potential opportunistic pathogens and “less likely uncertain pathogens” belonged to the genera *Aspergillus, Paecilomyces, Rhizopus, Chaetomium, Dichotomopilus, Paecilomyces* and *Trichoderma*. All tested *Aspergillus, Chaetomium, Dichotomopilus* and *Paecilomyces* isolates grew at neutral pH at 30 °C or 37 °C in contrast to most of the *Trichoderma* isolates, which were unable to grow at neutral pH at the tested temperatures. Most of the tested *Rhizopus* isolates were also unable to grow at 37 °C (Table 1).

### 3.2. Identification to Species Level of Potentially Pathogenic Indoor Isolates Assigned to Risk Group 1 and 2 Organisms

For the identification of isolates to the species level and classification to appropriate risk group, DNA-based methods are required [30,31]. Diversity tracking of the 284 isolates assigned them into 26 morphotypes included in 4 categories. Fungal isolates fulfilling the above criteria for pathogenicity were separated into 13 morphotypes (MT1–MT13) representing isolates with strong or moderate pathogenic potential (category A and B isolates in Table 1) and possible but uncertain pathogens MT14–MT19 (category C isolates in Table 1). Category D isolates included the isolates lacking pathogenic potential (MT20–MT26), mainly represented by *Penicillium, Rhizopus* and *Trichoderma* isolates unable to grow at neutral pH at temperatures > 30 °C. Selected isolates representing the genera *Aspergillus*, *Paecilomyces, Chaetomium, Dichotomopilus, Paecilomyces, Rhizopus,* and *Trichoderma* were identified to the species level. The results in Table 1 and 2 show that the morphotype-based identification to the genus level was confirmed by the identification of species. However, pathogenic potenial was detected in *Paecilomyces* sp. strains not identifiable to any known species by DNA sequencing of the marker genes used (Table 2).

Category A and B strains identified as *A. flavus, A. niger, A. tubingensis, A. calidoustus,* and *A. fumigatus* as well as *R. arrhizus, T. longibrachiatum* and *P. variotii* are well known as toxigenic and pathogenic species, representing risk group 2 organisms as defined by the European Parlament (2000) Directive 2000/54/EC. [1,2,12,13,32,33]. Interestingly, the *A. flavus* species contained two morphotypes differing in their ability to grow at pH 7.2 and 37 °C, possibly also reflecting a difference in pathogenic potential. Interestingly, *Aspergillus pseudoglaucus* tested positive for high pathogenic potential (=category A organism) but belongs to risk group 1 according to the European Parlament (2000) Directive 2000/54/EC, [32] and is considered as a rare and uncertain human pathogen, mainly causing superficial infections [2]. Strains of *T. citrinoviride* belonging to the Section *Longibrachiatum* were isolated from clinical samples and classified as a human pathogen [15,34].

The category C strains identified as *A. westerdijkiae*, *A. versicolor*, *C. globosum* and *C. cochliodes* represented risk group 1 organisms according to Directive 2000/54/EC [32] and include species known to cause superficial infections, onychomycoses and rare invasive infections in immunocompromised patients [4,11,12]. Interestingly, strains of the risk group 1 species *T. atroviride* tested negative for growth in neutral pH, indicating a lack of pathogenic potential, even though human infection by *T. atroviride* has been described [13,14]. The *D. finlandicus* isolates reported to grow at 37 °C on potato dextrose agar (pH 5–6) [25] did not grow on TSA or blood agar (BA) at 37 °C, but exhibited weak growth at 30 °C. This indicates that the possible pathogenicity of *D. finlandicus* cannot be excluded.

### 3.3. Strains of Paecilomyces Representing Possible New Species of Potentially Pathogenic Fungi

*Paecilomyces variotii* is a known pathogenic species [33]. It is interesting that the *Paecilomyces* sp. strains separated in morphotypes differed from *P. variotii* and were identified to the species level by the DNA sequence analysis of the target gene used (Table 1 and Table 2). The *Paecilomyces* sp. isolates detected in three buildings were all potentially pathogenic but exhibited three different morphotypes according to toxicity profile, indicating that they may represent different species or strains of one species producing different metabolites.

### 3.4. The Aspergillus Species in Buildings May Adapt to Biocides and Cleaning Chemicals

The isolated potentially pathogenic strains represented the common global building mycobiota dominated by the xerophilic genus *Aspergillus* [7]. Pathogenic *A. niger* and *A. flavus* strains were isolated from 18 and 8 buildings, respectively. *Aspergillus* was the dominant and most diverse of the pathogenic genera occurring in both “problematic” and “ordinary” buildings. All tested *Aspergillus* strains grew in neutral pH at ≥30 °C and exhibited at least possible pathogenic potential (Table 1 and Table 3).

The results from this study (Figure 1 and Figure 2, Table 4) may also indicate that when growing on solid medium and exposed to dehydration during a long incubation time, the tested *A. niger* strain benefited from the tenside used in cleaning chemicals, Genapol-X-080, at the expense of *Paecilomyces* sp. and *Chaetomium* sp. strains. Figure 2 indicates that at Genapol concentrations as low as 10 µg mL^−1^, the *A. niger* strain outcompeted the two other strains, and that the increased fitness provided by Genapol-X-080 to the *A. niger* strain was dose dependent.

Indoor strains of *A. niger, A. flavus, A. westerdijkiae, A. calidoustus* and *A. versicolor* were more resistant to biocides compared to other genera isolated from indoor environments [26]. This study shows that the tested *A. calidoustus* strain may exhibit selective advantage from the use of biocides (Table 5). The enhanced production of Hülle cells by a Triclosan-treated *A. calidoustus* strain (Table 5) may possibly illustrate enhanced survival on biocide-treated surfaces. Factors affecting the production of Hülle cells in *A. calidoustus* is not well known, but our results indicate that certain biocides may stimulate Hülle cell formation, which may increase resilience under the hard and destructive conditions provided by the indoor environment exposed to biocides and cleaning chemicals [35,36]. However, these single experiments illustrated in Figure 1 and Figure 2 and Table 4 and Table 5 are preliminary; the results only indicate that the effects of biocides and cleaning chemicals on the indoor microbiota is worth investigation.

Potentially pathogenic fungi are ubiquitous organisms and cannot be completely avoided in outdoor or indoor environments [10,37]. In low amounts and as a minor component of the diverse environmental microbiota, they increase resilience towards environmental microbial challenges [38]. However, the immense use of cleaning agents, including tensides, disinfectants and biocides during the COVID-19 pandemic may, at least in theory, have unexpected consequences for the indoor microbiota and may coincide with decreased resilience in exposed persons. It may be speculated that the use of biocides may affect virulence and favor the competition and/or survival of certain pathogenic *Aspergillus* species [35,36,38,39]. In such cases viable conidia of virulent strains of pathogenic *Aspergillus* species may possibly expose vulnerable persons in quantities exceeding their resilience and pose a potential risk for indoor air safety.

## 4. Conclusions

Buildings considered both as “ordinary” and as “problematic” contained potentially pathogenic fungi. Screening for pathogenic potential was a successful method for the detection and diversity tracking of potentially pathogenic indoor isolates. This study includes a description of possible pathogenic potential in new species belonging to the genus *Paecilomyces*. The genus *Aspergillus* represented by well-known pathogenic species was the dominant and most diverse genus found in buildings. It is tempting to speculate that tolerance to biocides and tensides in cleaning chemicals and building materials may contribute to resilience and enhanced virulence of pathogenic *Aspergillus* species adapted to indoor environment. Screening and identification of new emerging potentially pathogenic species from buildings is a topic for future research.

## 5. Materials and Methods

### 5.1. Experimental Design

Fungal isolates (284 strains) collected from 20 buildings were screened for pathogenic potential. The experimental design of the study is shown in Figure 3.

### 5.2. Strain Collection of 284 Fungal Isolates from 20 Buildings

The 284 isolates were originally isolated on MEA (malt extract agar provided by several manufacturers, such as a.o. Oxoid Ltd, Thermo Fisher, Heysham, Ireland; Sharlab, Barcelona, Spain) incubated for 3 weeks at room temperature. The isolates were randomly deposited into the collection without any preference for potential pathogens. Strains were included in the collection based on meeting at least one of the following three criteria: (1) they were recognized as the dominant morphotype of the sample; (2) they were screened positive for toxicity towards boar sperm and/or in somatic cell lines; (3) they were recognized as new species or species not earlier detected in indoor environments.

The isolates emanated from settled indoor dust, from dust collected from exhaust and inlet air filters and from building materials from 20 buildings. The total of 19 buildings represented 16 public buildings, such as educational buildings and sport facilities and offices and 3 apartments in Finland, while 2 isolates were derived from an educational building in Copenhagen, Denmark. Sixteen buildings were classified as “problematic buildings” based on reported moisture damage and/or indoor air quality complaints. Six buildings were classified as “ordinary buildings” not associated with moisture or complaints of any kind (yet).

### 5.3. Determination of Morphotypes

The criteria for diversity and assignation to morphotypes were revealed by 7 characteristics: by light microscopy (conidiophores, ascomata and sporangia, presence of Hülle cells), colony morphology on MEA at pH 5.4, fluorescence emission of biomass dispersals, pathogenic potential, i.e. the ability to grow in pH 7.2 at 30 °C and 37 °C, and toxigenicity as indicated by responses in two bioassays, boar sperm motility inhibition assay (BSMI) and inhibition of cell proliferation (ICP) performed as described previously [19,22,26]. A colony was considered toxic in the BSMI assay when <2.5 vol% of its biomass suspension inhibited the boar sperm motility after 30 min to 3 days of exposure. A colony was considered toxic in the in vitro ICP assay when < 5 vol % of its biomass suspension inhibited proliferation of the porcine kidney (PK-15) cells after 2 days of exposure. In both bioassays, a nontoxic strain *Trichoderma reesei* DSM 768 and a trilongin-producing *T. longibrachiatum* strain SzMC THG were used as nontoxic and toxic reference strains, respectively. Commercial mycotoxins used as references from Sigma-Aldrich (St. Louis, MO, USA) were sterigmatocystin, chaetoglobosin A (toxic positive controls) and ochratoxin A (negative control, not visible in the bioassays).

### 5.4. Screening for Pathogenic Potential

The pathogenic potential of the strains was tested by culturing on media with neutral pH at 37 °C and 30 °C. The plates used were tryptic soy agar (TSA), pH 7.2 (Oxoid Ltd., Thermo Fisher, Heysham, Ireland,) and BA (pH 7.2), Tammer Biolab, Tampere Finland). Reference plates used were malt extract agar (MEA), pH 5.5 (Sharlab, Barcelona, Spain) incubated at 22 °C. The plates were sealed with gas-permeable tape and incubated for 1 week.

The readout of the test was conducted as follows: If the colony growing on the TSA or BA plates at 37 °C had the same size and same degree of sporulation as that obtained on reference plates, then the readout of pathogenic potential was written as ++. If the colony size was smaller, exhibited weak or no sporulation compared to that on the reference plates, but the area of the colony had visibly enlarged after 7 d of incubation, then the readout for pathogenic potential was written as +. Isolates unable to grow at 37 °C were then tested for growth on TSA plates at 30 °C. If the colony grew at 30 °C on TSA or BA, then the readout of pathogenic potential was written as ±. For isolates unable to grow on TSA at 30 °C and on MEA at 37 °C, then the readout for pathogenic potential was written as -.

Based on the ability to grow on the test plates, TSA and/or BA at 37 °C and 30 °C, the tested isolates were grouped into 4 categories (A–D): (A) isolates with high pathogenic potential, (growth on test plates marked as ++); (B) isolates with moderate pathogenic potential (growth on test plates marked as +); (C) isolates with possible but uncertain pathogenic potential, (growth on test plates marked as ±); (D) isolates with no pathogenic potential (growth on test plates marked as -).

### 5.5. HPLC-MS Analysis and Identification of Mycotoxins

The collected fungal biomasses were extracted with ethanol, and fungal toxins were analyzed and identified by high-performance liquid chromatography–high-resolution mass spectrometry using an Agilent 6530C Q-TOF mass spectrometer (Agilent Technologies, Santa Clara, CA, USA) with a Dual Jet Stream electrospray ionization (ESI) source and an Agilent 1260 Infinity II series LC with DAD UV detector. The column used was a SunFire C18, 2.1 × 50 mm, 2.5 μm (Waters, Milford, MA, USA) using gradient elution solution A: H_2_O with 0.1% (*v*/*v*) formic acid and B: methanol. Separation was performed using the following gradient: 60–80% B at 0–15 min; 80–100% B at 15–20 min and 100% B at 20–40 min at a flow rate of 0.2 mL min^−1^. The mass spectrometer was operated in positive ion mode. The scanning range was 50–2300 m/z. The drying gas flow rate and temperature were set at 10 L min^−1^ and 325 °C. The sheath gas flow rate and temperature were set at 12 L min^−1^ and 400 °C, and the capillary voltage was 4000 V.

Ochratoxin in the ethanol extracts of the *Aspergillus westerdijkiae* strain PP2 was indicated by emitted fluorescence (ex 375 nm, em 426 nm) measured with a microplate reader (Fluoroskan Ascent, Thermo Scientific, Vantaa, Finland), commercial ochratoxin A (Sigma-Aldrich, St. Louis, MO, USA) was used as reference.

### 5.6. Identification of the Potential Fungal Opportunistic Pathogens to the Species Level

Selected isolates representing the potentially and possibly pathogenic morphotypes (MT1-MT13, and MT14-MT19) were identified by DSMZ (German Collection of Microorganisms and Cell Cultures GmbH, Braunschweig, Germany) or by amplification of the ITS, *tef1α*, [24] *CaM* [25] and/or *rbp2* [40] markers and their sequence analysis by Nucleodide BLAST (https://blast.ncbi.nlm.nih.gov/Blast.cgi accessed on 6 October 2022). The identifications were validated by pairwise BLAST analyses to compare the sequences with the reference sequences of the respective type strains of *Aspergillus* [41], *Paecilomyces* [41], *Rhizopus* [42], *Trichoderma* [43] and *Chaetomium* [44] species.”

### 5.7. Testing the Effects of Biocides and a Tenside on an Aspergillus calidoustus Strain

The effects of 2 biocides and a tenside on an *A. calidoustus* strain after dehydration due to long incubation were determined as follows: The tests were performed in malt extract broth (MEB, Oxoid, Hampshire United Kingdom) 12 g in 1 L, pH 5.5, glucose 7.1 mmol L^−1^, in microtiter plates with 96 wells. The biocides dissolved in ethanol and the tenside dissolved in water were diluted by two-fold dilutions in 100 µL MEB. Fungal spore suspensions, 100 µL PBS containing 10^6^ spores mL^−1^, were added to each well. The effects of the chemicals on spore germinantion, hyphal growth, resporulation of new conidia and occurrence of Hülle cells were inspected by a phase-contrast microscope (Olympus CKX41, Tokyo, Japan; magnification 400×) and an image recording software (Cellsense^®^ standard version 11.0.06, Olympus Soft Imaging Solutions GmbH, Münster, Germany) after 3 days and 9 weeks of incubation at 24 °C. The biocides and the tenside were purchased from Sigma-Aldrich (St. Louis, MO, USA). The selected biocides were Borax (600 μg mL^−1^; sodium tetraborate, CAS: 1330-43-4), Triclosan (5-chloro-2-(2,4-dichlorophenoxy) phenol), CAS: 3380-34-5) and the tenside and wetting agent Genapol X-080 (CAS: 9043-30-5).

### 5.8. Testing Genapol-X-080 on the Competitiveness of an Aspergillus niger Strain

The competitiveness test on solid media used on the *Aspergillus niger* strain A21 was performed by filling the wells of the microtiter plate with MEA (Oxoid Ltd, Thermo Fisher, Heysham, Ireland), 200 µL supplemented with 50 µL of PBS Genapol dilutions, 300 µg mL^−1^, 5 µg mL^−1^, and 50 µL spore suspensions, 10^6^ spores per ml of indoor isolates of *A. niger*, *C. cochliodes* and *Paecilomyces* sp. The plate was incubated for 20 d at room temperature, the lid was opened every third day and the plate was photographed. After the last day, the fungi growing in the wells were identified by microscopic inspection of conidiophores and ascomata.

## Figures and Tables

**Figure 1 pathogens-11-01171-f001:**
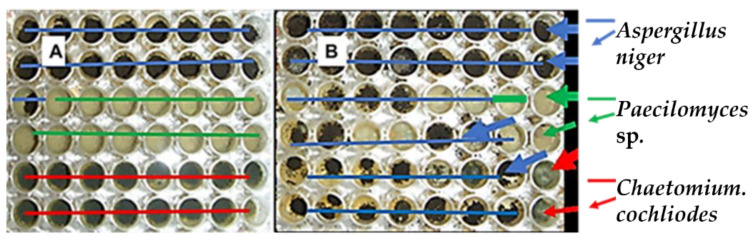
Difference in the distribution of *Aspergillus niger* strain Asp21 compared to *Paecilomyces* sp. strain Pec/skk and *Chaetomium cochliodes* strain CH2 in the absence (**A**) and presence (**B**) of Genapol. Both plates containing 200 µL of MEA were inoculated with the 3 strains, 16 wells per strain. Panel A shows the 48 wells with no Genapol addition. Panel B shows the plate, where 5 µL of twofold dilutions of Genapol-X-080 was added to each well, from 300 µg mL^−1^ in the first vertical row of wells to 4.7 µg mL^−1^ in the seventh and 0 µg ml^−1^ in the last vertical row of wells. The plates were imaged after 20 d of incubation at 22 °C.

**Figure 2 pathogens-11-01171-f002:**
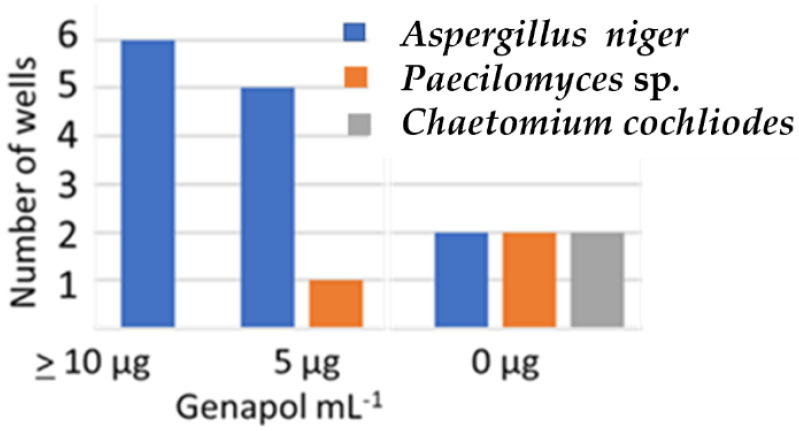
Distribution of three fungal isolates, *Aspergillus niger* Asp 21, *Paecilomyces* sp. Pec/skk and *C. cochliodes* CH2 in 18 microtiter plate wells containing MEA (200 µL) with Genapol additions of 10 µg mL^−1^, 5 µg mL^−1^, and 0 µg mL^−1^ (no Genapol added).

**Figure 3 pathogens-11-01171-f003:**
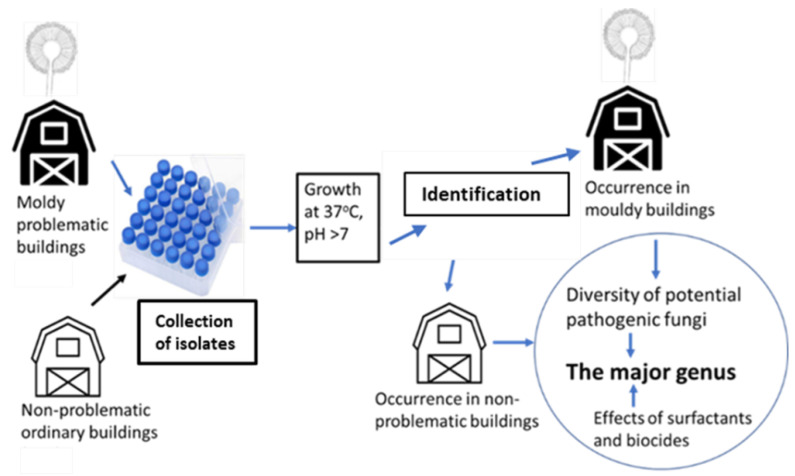
The experimental design of screening a strain collection of indoor fungi for pathogenic potential.

**Table 1 pathogens-11-01171-t001:** Morphotypes of 284 indoor fungal isolates from 20 buildings, 14 problematic (P) and 6 ordinary (O), non-problematic.

MorphologyMicroscopy (M)Colony MEA (C) Fluorescence of Cell Dispersal (F)	Growth at 37 °CpH 7.2	Toxicity Profile	Conidiospore/SporangiosporeSize in µm,Hülle Cells	From Problematic/Ordinary Building P/O
M	C	F		BMSI	ICP		P	O
A. Isolates with high pathogenic potential (growing well at 37 °C in pH 7.2)
1. Genus *Aspergillus*
MT1 (5 strains)
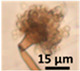	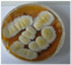		++	+	+	3–4 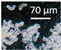	+	-
MT2 (1strain)
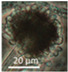	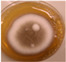		++	+	+	3–4	+	-
MT3 (5 strains)
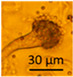	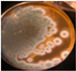		++	-	+	2–3	+	+
MT4 (21 strains)
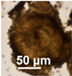	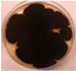		++	-	+	4–5	+	+
MT5 (2 strains)
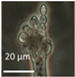	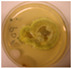		++	+	+	5 × 8	+	--
MT6 (10 strains)
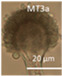	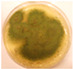		++	-	-	5–8	+	+
2. Genus *Rhizopus*
MT7 (4 strains)
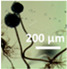	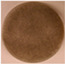		++	-	-	8–10	+	-
3. Genus *Trichoderma*
MT8 (6 strains)
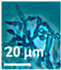	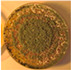		++	+	+	2 × 4	+	-
B. Isolates with moderate pathogenic potential(weak growth at 37 °C and pH 7.2)
1. Genus *Aspergillus*
MT9 (1 strain)
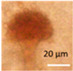	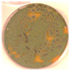		+	-		4–5	-	+
2. Genus *Paecilomyces*
MT10 (3 strains) Toxigenic isolates
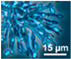	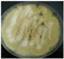		+	+	-	3 × 8	+	-
MT11 (2 strains) Toxigenic isolates
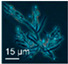	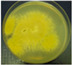		+	-	+	3 × 8	+	-
MT12 (2 strains) Non-toxigenic *Paecilomyces*
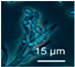	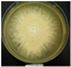		+	-	-	3 × 8	+	-
3. Toxigenic *Trichoderma*
MT13 (5 strains)
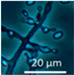	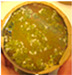		+*	+	+	1.5 × 3	+	-
C. Isolates with less likely pathogenic potential (not growing at 37 °C, but growing at 30 °C at pH 7.2)
1. Toxigenic *Aspergillus*
MT14 (12 strains)
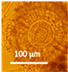	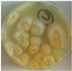		±	+	+	3-–4	+	+
MT15 (12 strains)
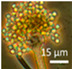	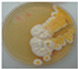		±	-	+	3–4	+	-
2. Toxigenic *Chaetomium*-like strains
MT16 (10 strains)
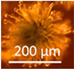	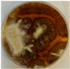		±	+	+	9–10	+	+
MT17 (4 strains)
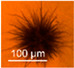	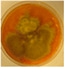		±	+	+	8 × 9	+	-
MT18 (6 strains)
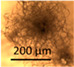	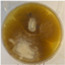		±	-	+	5 × 11	+	-
MT19 (3 strains)
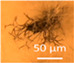	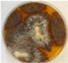		±	(+)	-	4 × 6	+	+
D. Isolates with no pathogenic potential (not growing at 37 °C and/or not growing at pH 7.2 at 30 °C)
1. Toxigenic *Acrostalagmus*
MT20 (8 strains)
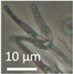	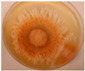		-	-	+	3 × 5	+	-
2. Toxigenic *Penicillium*
MT21 (8 strains
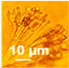	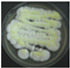		-	-	+	3–4	+	+
MT22 (10 strains)
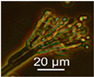	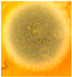		-	-	+	3–4	+	+
MT23 (6 strains)
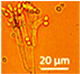	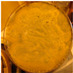		-	+	+	3–4	+	-
3. Toxigenic *Trichoderma*
MT24 (40 strains)
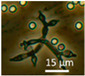	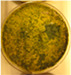		- **	+	+	3–5	+	-
4. Non-toxic *Penicillium*
MT25 (63 strains)
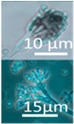	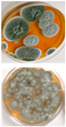	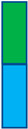	-	-	-	3–5	+	+
5. Non-toxic *Rhizopus*
MT26 (35 strains)
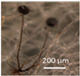	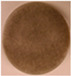		-	-	-	8–10	+	+

* Growth at 37 °C: ++ grew very well on MEA, + weak growth and no conidia on TSA. **—Did not grow on malt extract agar (MEA) medium at 37 °C, nor on TSA at 30 °C.

**Table 2 pathogens-11-01171-t002:** The 67 strains identified to species level representing the 19 morphotypes with pathogenic potential (MT1–MT13) and with possible pathogenic potential (MT14–MT19).

Morphotype	Species	Strain ID	Location of Isolation		GenBank Accession Number				
				Toxin/Metabolite	*tef1α*	ITS	*rpb2*	*CaM*	Ref.
**MT1**	*Aspergillus calidoustus*	MH4	Indoor settled dust, university office, Finland	Ophiobolins		KM853016			[19]
	*Aspergillus calidoustus*	MH21	Indoor settled dust, university office, Finland	ND					[19]
	*Aspergillus calidoustus*	MH34	Indoor settled dust, university office, Finland	ND					[19]
	*Aspergillus calidoustus*	MH36	Indoor settled dust, university office, Finland	ND					[19]
**MT2**	*Aspergillus calidoustus*	Asp16/LKK	Indoor dust, school, Finland	Ophiobolin K Ophiobolin G				OP356693	recent study ^1^
**MT3**	*Aspergillus fumigatus*	AE1	Indoor settled dust, gym, Finland	ND				OP295388	recent study
	*Aspergillus fumigatus*	AE5	Indoor settled dust, gym, Finland	ND				OP295389	recent study
	*Aspergillus fumigatus*	AH3	Indoor settled dust, gym, Finland	ND				OP295390	recent study
**MT4**	*Aspergillus niger*	Asp21	Exhaust filter school Finland	ND				OP356697	recent study
	*Aspergillus niger*	Asp1	Exhaust filter, school, Finland	ND				OP295392	recent study
	*Aspergillus niger*	AE4	Indoor settled dust gym Finland	ND				OP295391	recent study
	*Aspergillus tubingensis*	Asp2	Exhaust filter school Finland	ND				OP295394	recent study
	*Aspergillus tubingensis*	ASN	Indoor settled dust gym Finland	ND				OP295393	recent study
**MT5**	*Aspergillus pseudoglaucus*	8/SL	Fall out plate appartment Finland	ND				OP356700	recent study
**MT6**	*Aspergillus flavus*	7D	Extract filter school Finland	ND				OP356699	recent study
	*Aspergillus flavus*	1/37	Extract filter school Finland	ND				OP356698	recent study
	*Aspergillus flavus*	AGE	Indoor settled dust gym Finland	ND				OP295387	recent study
	*Aspergillus flavus*	AE2	Indoor settled dust gym Finland	ND				OP295385	recent study
	*Aspergillus flavus*	AEH	Indoor settled dust gym Finland	ND				OP295386	recent study
**MT9**	*A. flavus*	ASpf	Indoor settled dust gym Finland	ND				OP356696	recent study
**MT10**	*Paecilomyces variotii*	Paec2/kop = Pa/2 = Paec2 = HAMBI 3342 *	Indoor settled dust, university, Denmark	Viriditoxin					[20,21]
	*Paecilomyces variotii*	Paec1/kop *	Indoor settled dust, university, Denmark					DSMZ	recent study
**MT11**	*Paecilomyces* sp.	ST28	Settled dust, school, Finland	ND			OP356689	OP295396	recent study
	*Paecilomyces* sp.	ST32	Settled dust, school, Finland	ND			OP356690	KP889008	recent study
**MT12**	*Paecilomyces* sp.	Pec/hiss	Settled dust, university, Finland	ND				OP356695	recent study
	*Paecilomyces* sp.	Pec/skk	Extract air filter, school, Finland	ND				OP356694	recent study
**MT7**	*Rhizopus arrhizus*	M1/KI	Extract air filter, school Finland	ND		OP288193			recent study
**MT13**	*Trichoderma citrinoviride*	SJ40	Settled dust, office, Espoo, Finland	Trilongins	MH177004	KP889007			[22]
	*Trichoderma citrinoviride*	T4//LKK	Dust, school, Finland	ND		OP351639			[23]
**MT8**	*Trichoderma longibrachiatum*	T37/skk	Extract air filter, school, Finland	ND		OP345956			recent study
	*Trichoderma longibrachiatum*	Thb	Moisture-damaged residence, Finland	Trilongins		HQ593512			[24]
	*Trichoderma longibrachiatum*	Thd	Moisture-damaged residence, Finland	Trilongins		HQ593513			[24]
	*Trichoderma longibrachiatum*	Thg	Moisture-damaged residence, Finland	Trilongins	EU401624	EU401573		EU401492	[24,25]
**MT14**	*Aspergillus westerdijkiae*	PP2 *	Vacuum cleaner dust, apartment, Finland	Stephacidin Avrainvillamide Ochratoxin					[24]
	*Aspergillus westerdijkiae*	KaIII *	Fall out plate, kindergarten, Finland	Stephacidin Avrainvillamide					[24]
	*Aspergillus westerdijkiae*	PP31 *	Vacum cleaner dust, apartment, Finland	ND					[24]
	*Aspergillus westerdijkiae*	PP3 *	Vacum cleaner dust, apartment, Finland	ND					[24]
	*Aspergillus westerdijkiae*	PP31 *	Vacum cleaner dust, apartment, Finland	ND					[24]
**MT15**	*Aspergillus versicolor*	SL/3 *	Settled dust, university office, Finland	Averufin Sterigmatocystin 5-Methoxysterigmatocystin					recent study ^1^
	*Aspergillus versicolor*	GAS226 *	Settled dust, office, Finland	Averufin Sterigmatocystin 5-Methoxysterigmatocystin					recent study ^1^
	*Aspergillus versicolor*	K20	Settled dust, university Finland	Sterigmatocystin					[19]
	*Aspergillus versicolor ***	MH10	Settled dust, university Finland	ND					[19]
	*Aspergillus versicolor ***	MH11	Settled dust, university Finland	ND					[19]
	*Aspergillus versicolor ***	MH25	Settled dust, university Finland	ND					[19]
	*Aspergillus versicolor ***	MH26	Settled dust, university Finland	ND					[19]
	*Aspergillus versicolor ***	MH32	Settled dust, university Finland	ND					[9]
	*Aspergillus versicolor ***	MH33	Settled dust, university Finland	ND					[9]
	*Aspergillus versicolor ***	MH35	Settled dust, university Finland	ND					[19]
**MT16**	*Chaetomium globosum*	C13/LM	Exhaust air filter, school, Finland	ND	MW556666				[26]
	*Chaetomium globosum1*	C22/LM	Exhaust air filter, school,	Chaetoglobosin Chaetoviridin A Chaetomugilin D Chaetoviridin C	MT498109				recent study ^1^
	*Chaetomium globosum*	MH5	Settled dust, public building, Finland	Chaetoglobosin Chaetoviridin A Chaetoviridin C	MT498108				[26]
	*Chaetomium globosum*	MÖ9	Settled dust, piggery, Finland	ND	MT498106	MW541924			[26]
	*Chaetomium globosum*	2c/26	Settled dust, apartment, Finland	ND	MW310244				[26]
	*Chaetomium globosum*	2b/26	Settled dust, apartment, Finland	ND	MT498110				[26]
	*Chaetomium globosum*	C22	Settled dust, apartment, Finland	ND	MW556668				[26]
	*Chaetomium globosum*	MH52	Settled dust, public building, Finland	ND	MT498107				[25]
	*Chaetomium globosum*	Ruk10	Settled dust, apartment, Finland	Chaetoglobosin Chaetoviridin A Chaetoviridin C	MT498101	MW541927			[26,27]
	*Chaetomium globosum*	MTAV35*	Settled dust, university, Finland	Chaetoglobosin Chaetoviridin A Chaetoviridin C					[26,27]
	*Chaetomium globosum*	3b/APP	Exhaust air filter, public building, Finland	ND	MW588207				[26]
**MT17**	*Chaetomium cochliode*	CH2	Indoor stettled dust, gym Finland	Chaetomin Chaetomuglin C Chaetoviridin A			OP356691	OP295395	recent study ^1^
	*Chaetomium cochliodes*	CH3	Indoor stettled dust, gym, Finland	ND			OP356692		recent study
	*Chaetomium cochliodes*	OT7	Settled dust, office, Finland	Chaetomin Chaetoviridin A Chaetomugilin D	MT498103				[26,27]
	*Chaetomium cochliodes*	OT7B	Settled dust, office, Finland	Chetomin Chaetoviridin Chaetomugilin D	MT498102				[26,27]
**MT18**	*Chaetomium rectangulare*	MO13	Settled dust, piggery, Finland	ND	MT498104 MT498105	MW541928 MW541929			[26]
**MT19**	*Dichotomopilus finlandicus*	Ch1/Tu	Inlet air filter, public building, Finland	ND	MT644127	MW541926	MZ665531		[26,27]
	*Dichotomopilus finlandicus*	C5/LM	Exhaust air filter, school, Finland	ND	MW556671	MW541925	MZ665530		[26]

*** identified at the German Collection of Microorganisms and Cell Cultures GmbH (DSMZ). ** Identified as *Aspergillus versicolor* based on morphology, responses in two bioassays and fluorescence emission and similarity to the strains 3/SL and GAS226 identified as *A. versicolor* by DSMZ [19]. ND = Not determined. ^1^ Monoisotopic mass ions of the identified compounds in this study: Ophiobolin K: [M + H]^+^ at *m/z* 385.2719, [M + Na]^+^ at *m/z* 407.2541 and [2M + Na]^+^ at *m/z* 791.5230; Ophiobolin G: [M + H]^+^ at *m/z* 367.2640, [M + Na]^+^ at *m/z* 389.2893 and [2M + H]^+^ at *m/z* 733.5117; Averufin: [M + H]^+^ at *m/z* 369.0972, [M + Na]^+^ at *m/z* 391.0797; Sterigmatocystin: [M + H]^+^ at *m/z* 325.0718, [M + Na]^+^ at *m/z* 347.0536; 5-Methoxysterigmatocystin: [M + H]^+^ at *m/z* 355.0819, [M + Na]^+^ at *m/z* 377.0639; Chaetoglobosin: [M + H]^+^ at *m/z* 529.2691, [M + Na]^+^ at *m/z* 551.2521; Chaetoviridin A: [M + H]^+^ at *m/z* 433.1416, [M + Na]^+^ at *m/z* 455.1244, [2M + Na]^+^ at *m/z* 887.2584; Chaetomugilin D: [M + H]^+^ at *m/z* 435.1566, [M + Na]^+^ at *m/z* 457.1381, [2M + Na]^+^ at *m/z* 891.2885; Chaetoviridin C: [M + H]^+^ at *m/z* 435.1581, [M + Na]^+^ at *m/z* 457.1399, [2M + Na]^+^ at *m/z* 891.2856; Chaetomin: [M + H]^+^ at *m/z* 711.1174, [M + Na]^+^ at *m/z* 733.0996, [2M + Na]^+^ at *m/z* 1443.2105; Chaetomugilin C: [M + H]^+^ at *m/z* 433.1411, [M + Na]^+^ at *m/z* 455.1233, [2M + Na]^+^ at *m/z* 887.2574.

**Table 3 pathogens-11-01171-t003:** Diversity of potentially pathogenic fungal isolates representing common indoor genera from 20 buildings. Fungi with names set in bold were isolated from both problematic and non-problematic buildings. Fungi with names set in regular font were from problematic buildings only.

Total Isolates *n* = 296	Found in nr of Buildings *n* = 20	Potentially Opportunistic Pathogenic and Less Likely Pathogenic isolates *n* = 114	Found in nr of Buildings *n* = 19	Growth at pH 7.2	
				37 °C	30 °C
*Aspergillus*	18	*A. calidoustus*	2	+	+
		** *A. flavus* **	8	+	+
		** *A. fumigatus* **	4	+	+
		** *A. niger* **	18	+	+
		** *A. tubingensis* **	2	+	+
		** *A. westerdijkiae* **	17	-	+
		*A. pseudoglaucus*	2	+	+
		*A. versicolor*	6	-	+
***Chaetomium*-like**	9	** *C. globosum* **	6	-	+
		*C. cochliodes*	2	-	+
		*C. rectangulare*	1	-	+
		** *D. finlandicus* **	2	-	+
*Paecilomyces*	4	*Pae. variotii*	1	+	+
		*Paecilomyces* sp.	5	+	+
** *Penicillium* **	20	*Pen. chrysogenum*	2	-	
** *Rhizopus* **	20	*R. arrhizus*	1	+	+
*Trichoderma*	10	*T. citrinoviride*	2	+	+
		*T. longibrachiatum*	2	+	+
		*T. atroviride*	10	-	-

**Table 4 pathogens-11-01171-t004:** Occurrence of conidia of *Aspergillus niger* Asp 21, *Paecilomyces* sp. Pec/skk and *Chaetomium cochliodes* CH2 in wells of three microtiter plates incubated for 20 d.

Number of Wells (*n* = 48) with Conidia Recognized by Microscopy			
Genapol Addition			No Genapol
	Plate 1	Plate 2	Plate 3
*Paecilomyces* sp. strain Pec/skk	3	0	15
*Chaetomium cochliodes* strain CH2	2	0	16
*A. niger* strain Asp21/skk	41	34	17
No conidia	2	14	0

**Table 5 pathogens-11-01171-t005:** Effect of a tenside and two biocides and response to drying in the *Aspergillus calidoustus* strain MH 34. The spore suspension of the strain and the tenside and biocides were added to 200 µL of malt extract broth and imaged after 3 d and 9 weeks of exposure.

1. Effects of germination of conidia after 3 d *				
	Germinated conidia +++	Germinated conidia +++	Inactive conidia +++	Inactive conidia +++
	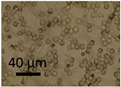	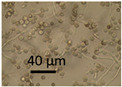	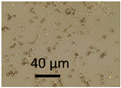	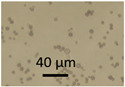
	Control	Genapol	Borax	Triclosan
		5000 µg mL^−1^	5000 µg mL^−1^	16 µg mL^−1^
2. Effects of biocides, tenside and drying for 9 weeks *				
Inactive conidia		Hyphens	New conidia	Hülle cells
µg mL^−1^	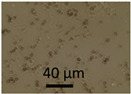	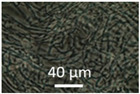	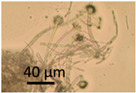	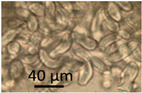
Borax 5000	+++	-	-	-
2500	-	-	++	+
1250	-	-	++	+
Triclosan				
16	+++	-	-	-
8	+	-	-	+++
4	-	+	+	+
Genapol				
5000	-	+++	-	-
1000	-	+++	-	-
100	-	-	++	-
3. Control, incubation for 9 weeks				
Inactive conidia		Hyphens	New conidia	Hülle cells
Control	-	-	+++	+

* The images represent the average view of five different fields. +++ means the view looked similar to the image, ++, the structures in the image were visible in half of the amount in the image, + the structures were visible in about 1/10 of the amount of the image means that the structures were not visible.

## Data Availability

Raw data are available upon request to the authors.

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
