# Peer review of "Aspergillus Was the Dominant Genus Found during Diversity Tracking of Potentially Pathogenic Indoor Fungal Isolates"

_pathogens, 2022, doi:10.3390/pathogens11101171_

Round 1
Reviewer 1 Report
In the manuscript entitled “Aspergillus was the dominant genus found during diversity tracking of potentially pathogenic indoor fungal isolates” submitted to Pathogens, the authors deal with the screening of 284 fungal strains, previously isolated in problematic and non-problematic building, to evaluate their pathogenicity. The submitted manuscript is generally interesting, fluent and well written. The data are clearly presented, and the manuscript features informative tables with beautiful pictures.
In the section 5.4 Screening for Pathogenic Potential:
L2-3 it would be advisable to write in extenso the medium name the first time
L5 “gas-permeable tape and incubated for 1 w.” I think the word week was cut off by mistake
Why the use of MEA for the reference plates, instead of the same media with buffered pH? Morphology and growth rate greatly vary according to different media composition…
Despite sufficient references are reported I would suggest you, in order to increase the readability, to add more details on the test to explain the procedures aimed at evaluating effects of biocides and Genapol .
In the Data Availability Statement I suggest to declare that the raw data are available upon request to the authors.
Author Response
The answers to reviewr 1 is attached

Reviewer 2 Report
This paper treats an interesting matter, but I found several issues that need to be corrected or changed until it can be envisaged to publish it.
Most of the methology seems to be state of the art. However, the wording should be revised carefully, and at least it should be clarified that most of the species that are mentioned here are not classified as pathogens, accoridng to international standards. Opportunistic pathogens are of course often encountered among the soil molds, but alone the ability to grow at 37°C is not a proof that the respective strains can attack humans.
I do not know these bioasssays too well but at least the cytoxox test the authors used is very sensitive and even extracts with large amounts of fatty acids will show positive results.
The detection of mycotoxins by HPLC/DAD-MS should be illustrated by images such as chromatograms and spectra in the SI at least for some examples.
The GenBank acc nos were not all accessible, and I recommend also to deposite the sequence data with the SI. If the authors used BLAST searches for comparison with type strains the corresponding references should be cited (and not just their own GenBank acc nos.
Further, minor comments are in the pdf.

Author Response
The answeres to the reviewer 2 is attached

Round 2
Reviewer 2 Report
I am now okay with the new version where most of my comments have been answered and the appropriate changes were made in the revised ms.